# Acromioclavicular Joint Lesions in Adolescents—A Systematic Review and Treatment Guidelines

**DOI:** 10.3390/jcm12175650

**Published:** 2023-08-30

**Authors:** Naman Wahal, Alper Sukru Kendirci, Carlos Abondano, Mark Tauber, Frank Martetschläger

**Affiliations:** 1Deutsches Schulterzentrum, ATOS Klinik, 81925 Munich, Germany; 2Department of Orthopaedics and Traumatology, Istanbul Faculty of Medicine, Istanbul University, 34093 Istanbul, Turkey; alpersukru@gmail.com

**Keywords:** acromioclavicular joint, pediatric, young, dislocation, treatment

## Abstract

True acromioclavicular joint (ACJ) injuries are rare in children and adolescents due to the strength of ligaments in this age group. However, a standardized management guideline for these injuries is currently lacking in the literature. This systematic review aims to provide an organized overview of associated injuries and propose a management algorithm for pediatric ACJ injuries. Using the PRISMA (Preferred Reporting Items for Systematic Reviews and Meta-Analyses) guidelines, a systematic review was conducted. Two independent observers searched PubMed, Cochrane Central Register of Controlled Trials (CENTRAL), and Scopus databases for ACJ injuries in children and adolescents. The extracted data were analyzed (due to the limited number of publications and inhomogeneity of data, no formal statistical analysis was conducted), and cases were categorized based on injury frequency and pattern, leading to the formulation of a treatment algorithm. The risk of bias was assessed using the Joanna Briggs Institute (JBI) critical appraisal checklist. A total of 77 articles were identified, and 16 articles (4 case series and 12 case reports) met the inclusion criteria. This study included 37 cases in 36 patients (32 males, 4 females) with a mean age of 13 years (9–17 years). Six injury categories were described. Surgical management was performed in 27 ACJ injuries (25 open, 2 arthroscopic). Various surgical implants were used including K wires, polydioxanone sutures (PDS), screws, hook plates, suture anchors, and suture button devices. Most cases achieved good to excellent outcomes, except for one case of voluntary atraumatic dislocation of the ACJ. This systematic review provides the first comprehensive analysis of ACJ injury management in adolescents with open physis. It categorizes injury patterns and presents a treatment algorithm to enhance the understanding of these injuries. The review’s findings contribute valuable insights for clinicians dealing with pediatric ACJ injuries.

## 1. Introduction

‘No impediment, small or great, would result from such an injury at the shoulder, only there will be a deformity at the shoulder’ [1]. Hippocrates believed this to be true centuries ago. These famous words by him, however, failed to stand the test of time. Various long-term complications of such injuries in children (such as duplication of the clavicle, chronic instability, cosmetic deformities and clavicle shortening) have been reported [2,3]. Anatomists and orthopaedic surgeons have constantly debated and struggled to understand the complex pathoanatomy, biomechanics and management of acromioclavicular joint (ACJ) injuries for as long as medical interventions have existed. But the guidelines for the management of such injuries continue to be inconsistent and somewhat controversial [4,5].

Eidman in his classic 1980 presentation at the American Orthopaedic Society for Sports Medicine in Big Sky, Montana, asserted that true AC joint injuries do not classically occur in children under 13 years of age; only pseudo-dislocations are seen under this age [6]. Recent studies clearly show that, although rare, true dislocations do occur in children from 11 up to 17 years of age [6]. A recent epidemiological study reported that only 50 AC joint dislocations in the pediatric population are treated on an inpatient basis in Germany every year [7]. This rarity is attributed to the strong protective ligaments and thick periosteum of the bones [8,9,10]. Accordingly, the literature on the diagnosis and management of such injuries in this population is sparse and comprises occasional case reports and small case series.

The current X-ray-based classification system (Dameron and Rockwood, Figure 1) is not only unable to provide a clear treatment guideline but also fails to include more common injuries like pseudo-dislocations of the AC joint (the apparent AC joint dislocation on radiographs in distal clavicle fractures) [6,8,9].

This article provides a first-ever systematic review of the published literature on the AC joint and associated injuries around the ACJ in the adolescent shoulder while simultaneously suggesting a management algorithm.

## 2. Materials and Methods

Preferred Reporting Items for Systematic Reviews and Meta-Analyses (PRISMA) guidelines were followed while conducting and reporting this systematic review [12].

### 2.1. Study Question

PICO (Population Intervention Outcome Control) criteria were used to construct the following study question: What is the management of ACJ injuries/dislocations (Intervention) in children and adolescents below the age of 18 years (Population) and how do the Outcomes compare with relevant Controls, if any? ACJ injuries were defined if the patient presented a clinical subluxation or dislocation of the ACJ.

### 2.2. Ethical Committee Clearance

Ethical committee clearance was not required as the review included data already published and available in the public domain.

### 2.3. Inclusion and Exclusion Criteria

Original studies reporting the management and outcomes following an intervention in ACJ injuries/dislocation in children and adolescents were included.

Studies in languages other than English, studies published before 2000 and studies reporting cases in patients above the age of 18 years were excluded. Studies that reviewed literature and did not contain original research were also excluded.

### 2.4. Literature Search

A comprehensive database search was performed by two independent researchers in PubMed (January 2000 to January 2021), Cochrane Central Register of Controlled Trials (CENTRAL) (January 2000 to January 2021) and Scopus (January 2000 to January 2021). Additional records identified via manual search from the bibliography of published studies were also included in the review. The keywords used were ‘Acromioclavicular’, ‘AC joint’; ‘injury’, ‘dislocation’, ‘subluxation’; ‘children’, ‘teenage’, ‘young’ and ‘adolescent’ in various combinations. Titles and abstracts were searched. The last search was performed on 1 May 2021.

### 2.5. Data Extraction

Data extraction was performed by one author and then checked by another. Disagreements were resolved via discussions among themselves (NW, ASK) and the senior author (FM).

The extracted data were then categorized based on the frequency and site of injury of the Superior Shoulder Suspensory Ligament Complex (SSSC). A management algorithm was developed after the analysis of the extracted data (taking into consideration the type of injuries encountered, interventions used and the outcomes reported in all the studies) and discussion with the senior author (FM).

### 2.6. Quality Appraisal and Bias Assessment

All the studies included were assessed using the Joanna Briggs Institute (JBI) critical appraisal checklist by two reviewers (NW, ASK) to check methodological quality and risk of bias [13,14]. Discrepancies were resolved via discussions among themselves.

## 3. Results

### 3.1. Study Selection

A total of 70 studies were selected from PubMed, Cochrane Central Register of Controlled Trials and Scopus. An additional manual search provided 7 studies. After removing the duplicates and other studies that did not fulfill the eligibility criteria, a total of 16 studies were selected for this review (Figure 2).

### 3.2. Study Characteristics

There were 12 case reports and 4 case series (Table 1). No randomized controlled trial (RCT) was found.

### 3.3. Quality Appraisal and Bias Assessment

All the included case reports and case series had a low risk of bias according to the JBI critical appraisal questionnaire (Table 2 and Table 3).

### 3.4. Patient Demographics (Table 1)

All the patients (32 males and 4 females) were children or adolescents under the age of 18 years; most of them suffered an acute traumatic injury to the acromioclavicular joint (ages 9 to 17). While the cause for ACJ injury was unilateral trauma for most patients, 2 females suffered from atraumatic voluntary ACJ instability. Only 1 female had atraumatic instability bilaterally [27].

### 3.5. Imaging (Table 1)

Plain Anteroposterior (AP) radiographs were the most commonly used first investigation. Special investigations like Magnetic Resonance Imaging (MRI), MRI arthrograms, and Computer Tomography (CT) scans were utilized in 9 studies. Kirkos et al. suggested an MRI in all cases to avoid overtreatment by surgery in stable cases [17].

### 3.6. Diagnosis (Table 1)

Out of the total number of 37 cases, only 7 cases of ACJ dislocation were isolated traumatic ACJ dislocation. Pseudo-dislocation (14 cases) and coracoid fracture with AC joint dislocation (10 cases) were the most frequent, and there was only 1 reported case of Triple injury in this age group. There were 7 true isolated AC joint dislocations, 3 cases of atraumatic voluntary dislocations and 2 cases of AC joint dislocation with concomitant distal clavicle fractures.

### 3.7. Intervention (Table 1)

A total of 27 ACJ injuries were managed surgically (25 ORIFs and 2 arthroscopic fixations), whereas the remaining 10 were managed conservatively.

All of the isolated AC joint dislocations (7 cases), clavicle fractures with ACJ dislocation (2 cases) and the Triple injury (1 case) were operated on. A total of 12 cases out of 14 pseudo-dislocations resulted in operations.

Operative management was decided based on injury to the capsule-ligamentous complex on imaging, presence of deformity and significant displacement, impingement of soft tissues and clinical judgment of the surgeon. The reason for non-operative management was young age with minor complaints or an intact capsule-ligamentous complex on imaging. Rockwood’s classification was not considered important while making management decisions and could only be applied in 7 cases (isolated true AC joint dislocations) out of 32 [7] (Table 1). Barchick et al. [27] did not comment on their choice of non-operative management in one shoulder and operative in the contralateral shoulder.

The most commonly used implant was K-wire (15 cases) followed by Hook-plate (4 cases) and polydioxanone sutures (PDS). Coracoid fixation when performed was achieved with screws. All cases were ORIF (Open Reduction and Internal Fixation) except for 2 arthroscopic surgeries with suture button fixation devices. Arthroscopic surgeries were performed on 15-year-old patients with no mention of the complications related to open physis (Table 1). Other implants used were cannulated screws and threaded pins.

### 3.8. Conservative Management (Table 1)

Out of the 10 cases of conservative management, 4 did not consent to surgery [22], 3 had minimally displaced stable injuries [15,21,23], 1 case of pseudo-dislocation was reduced successfully under sedation and 2 cases of voluntary dislocation were managed conservatively because of minor symptoms.

### 3.9. Outcome and Complications (Table 1)

A total of 4 studies reported on the existence of complications [24,26,27,28]. The outcome was good to excellent in all cases and in all categories except the voluntary ACJ dislocation, where both patients had poor outcomes (1 patient was managed surgically and the other conservatively) [26,27]. No major complications like infection or redislocation were reported. Scar complication and pseudo-bursa formation around the K-wire tip occurred in 2 patients [24,28].

### 3.10. Quantitative Synthesis of Data and Statistical Analysis

Out of 16 studies, 12 were case reports and 4 were small case series with less than 10 patients. Considering the low level of evidence and inhomogeneity of the data (population characteristics, diagnosis, imaging, intervention and outcome), formal statistical analysis and quantitative synthesis of data were not performed, as even major differences could not reach significance.

### 3.11. Categorization of the Injuries

The injuries were categorized based on the site of injury of the SSSC and organized as per the frequency of occurrence in the literature. A total of 6 categories are described from type 1 to type 6 in decreasing order of frequency of the injuries (Table 4, Figure 3). Accordingly, the maximum number of cases reported were grouped under type 1 (14 cases) and minimum (1 case) was grouped under type 6.


*Type 1: Pseudo-dislocation of the AC joint (Figure 3a)*


A total of 14 cases were reported, out of which 12 were operated (Table 1). Operative intervention was chosen in 12 cases due to significant displacement or soft tissue impingement [15,17,18,19,20,24].


*Type 2: Coracoid fracture with ACJ dislocation (Figure 3b)*


A total of 10 cases were reported, out of which 4 underwent surgery (Table 1) Mondori et al. and Jettoo et al. suggested operative intervention in all cases due to the instability of the SSSC caused by the combination of injuries [22,25]. Conservative management was conducted in minimally displaced injuries oH.


*Type 3: True dislocation of the ACJ (Figure 3c)*


All 7 cases (100%) reported in the literature resulted in operations (Table 1). The reasons for the operations in the published report were excessive displacement and trapezius muscle impingement [7].


*Type 4: Voluntary atraumatic dislocation of the ACJ (Figure 3d)*


Only 1 case out of 3 was operated on arthroscopically resulted in a poor outcome (Table 1). The reasons for conservative management were relatively minor complaints by the patient and avoidance of the inadvertent overtightening of ligament–capsular structures [26,27].


*Type 5: Distal Clavicle fracture with true ACJ dislocation (Figure 3e)*


Only 2 cases with true dislocation of the ACJ with distal clavicle fractures were reported and both were managed surgically due to the unstable injury pattern and shortening of the shoulder girdle [24,28].


*Type 6: “Triple Injury”: Coracoid fracture with pseudo-dislocation of ACJ with CC ligament rupture (Figure 3f)*


Only 1 case was reported in a child, and it was managed operatively (Table 1) as it was considered highly unstable [29].

### 3.12. Development of the Management Algorithm

A management algorithm was formulated utilizing the data extracted from published sources. Diagnostic modalities, management options and outcomes were systematically organized, which culminated in a coherent and detailed step-by-step algorithm for effectively addressing such cases (Figure 4).

## 4. Discussion

From the findings of this systematic review, the researchers successfully compiled and summarized the interventions while also developing a management algorithm (refer to Figure 4) for the cohort of adolescents with AC joint injuries (PICO). This review encompasses studies focusing on AC joint injuries within the adolescent age group starting from the year 2000. Notably, distinctive cases where ACJ injury is accompanied by concomitant SSSC injuries and pseudo-dislocation of the AC joint (which present clinically as true ACJ dislocation, unlike in adults where it is treated as a clavicle fracture) have been incorporated to provide a holistic overview of the entire spectrum of AC joint injuries within this specific age cohort. Nevertheless, the review did not identify any pertinent control groups among the studies incorporated in this analysis.

Although concrete management guidelines are lacking, most of the studies in this review chose operative management in cases with soft tissue impingement, buttonholing of the clavicle in the trapezius, excessive displacement of the clavicle and presence of concomitant injuries like coracoid fractures and clavicle fractures [7,16,18,19,22,24,28]. On the other hand, conservative management was suggested for minimally displaced dislocations and voluntary atraumatic dislocation due to their chronic nature and minimal symptoms [26,27]. A number of 4 cases with concomitant coracoid fractures were managed conservatively as the patients did not consent to surgery [22]. No case of isolated ACJ dislocation with minimal displacement was reported in the literature. Formal statistical analysis was not performed due to the inhomogeneity of the reported data and given the rarity of the injury. We were able to answer the study question (PICO) satisfactorily by providing a management algorithm for adolescent patients presenting with an ACJ injury.

### 4.1. The Existing Classification and Its Shortcomings

Dameron and Rockwood have described a classification system for pediatric ACJ injuries that is similar to the Rockwood classification for such injuries in adults [9]. The main pathoanatomic difference is intact coracoclavicular (CC) ligaments and the presence of thick periosteum that is disrupted only in higher-graded injuries (Figure 1).

It is based on 2-D X-ray images of a 3-D joint. It does not include common injuries like pseudo-dislocation and concomitant injuries like clavicle fractures and coracoid fractures. In this review, only 7 cases (isolated true AC joint dislocations) out of 32 (21.8%) could be classified according to this system. None of the published studies followed it to guide their treatment protocol.

### 4.2. The New Categorization of AC Joint Injuries According to Frequency of Occurence


*Type 1: Pseudo-dislocation of the AC joint (Figure 3a)*


Uncommon fractures of the distal clavicle, especially Salter-Harris type 1 or 2, can mimic AC joint dislocation due to physis involvement. The medial clavicular fragment typically slips through the torn periosteum while the AC joint and ligaments stay intact [16,29] (Figure 3a). Surgery (12 cases out of 14) was favored for significant displacement or soft tissue impingement while stable patterns with minimal displacement were managed conservatively (Table 1).


*Type 2: Coracoid fracture with ACJ dislocation (Figure 3b)*


This is a challenging diagnosis due to tissue overlap on radiographs. The SSSC is injured at two sites, justifying the recommendation for surgery by Mondori et al. and Jettoo et al. [22,25].


*Type 3: True dislocation of the ACJ (Figure 1 and Figure 3c)*


All 7 cases reported in the literature underwent surgery. Similar to the pseudo-dislocations (Type 1), the reasons for the operation in the published report were excessive displacement and trapezius muscle impingement [7].


*Type 4: Voluntary atraumatic dislocation of the ACJ (Figure 3d)*


Only 1 out of 3 cases underwent arthroscopic surgery, which resulted in a poor outcome (Table 1). It was the only category in which surgery yielded worse results than conservative management, which is attributed to the inadvertent overtightening of ligaments [26,27].


*Type 5: Distal Clavicle fracture with true ACJ dislocation (Figure 3e)*


Both reported cases were managed surgically (Table 1). Acharya et al. detailed a 12-year-old’s fracture fixed surgically with K-wire due to a disrupted SSSC ring at two points, rendering the shoulder girdle unstable [28].


*Type 6: “Triple Injury”: Coracoid fracture with pseudo-dislocation of ACJ with CC ligament rupture (Figure 3f)*


This highly unstable injury was managed successfully by Duerr et al. [29] using a cannulated screw for the coracoid fracture, transosseous sutures for ACJ stabilization and Suture Tak anchors (Arthrex, Naples, FL, USA) for CC ligament repair.

### 4.3. Management Algorithm (Figure 4)

The first investigation to confirm the diagnosis was the standard panoramic Zanca view with a contralateral AC joint. Open fractures, displacement larger than the diameter of the clavicle, soft tissue impingement or impending skin perforation, perforation of the trapezius muscle by the dislocated clavicle or fracture fragments, neurovascular impingement and severe shortening of the shoulder girdle are the absolute indications for surgical management [18,24]. The next step is to perform an MRI and assess the status of the SSSC. Disruption of the SSSC at two or more than two sites indicates instability and a severe injury that requires surgical stabilization (in accordance with the ‘Double Disruption of the SSSC’ theory by Goss [30]).

In terms of implants, the AC joint was fixed with K-wires in most of the studies (Table 1). However, using PDS in the tension band technique (Table 1) allows for second implant removal surgery in the follow-up period to be avoided. Although arthroscopic fixation using a suture button device showed promising results in a 15-year-old child, it provided undue stress on the growing coracoid physis [7,22]. The fixation of coracoid fractures was performed using cannulated screws in the majority of the reports (Table 1).

The term ‘AC joint injury equivalents’ is used to describe all the injury patterns that resemble AC joint dislocation in children and adolescents (other than the isolated true AC joint dislocations described by Dameron and Rockwood in their classification).

### 4.4. Limitations and Strengths

It is important to acknowledge that the inclusion of case reports and small case series in this review contributes to a relatively low level of evidence. The data presented in this study were found to be heterogeneous in terms of patient population, diagnosis, imaging modality, intervention and rationale for intervention. However, it is worth noting that all the studies included in this review received a JBI critical assessment score above six, indicating a relatively high methodological quality.

Furthermore, this review offers a comprehensive and exhaustive analysis of all English-language published reports from the year 2000, providing a complete picture of AC joint injuries in adolescents. This study represents the first review of its kind, allowing for a thorough understanding of these injuries based on the 3D pathoanatomy of the joint. In addition, the newly proposed management algorithm serves as a practical guideline for the treatment of this rare injury, providing valuable insights to clinicians and readers.

Further research with prospective case series or trials in this area is needed as it would benefit clinicians by providing a higher level of evidence for management.

## 5. Conclusions

ACJ injuries in children and adolescents are rare, with a limited number of admissions each year. In this study, we developed a comprehensive treatment algorithm based on published reports and a categorization system that considers the pathoanatomy of the shoulder joint. This algorithm aims to provide clinicians with a practical framework for delivering appropriate care to patients with ACJ injuries.

## Figures and Tables

**Figure 1 jcm-12-05650-f001:**
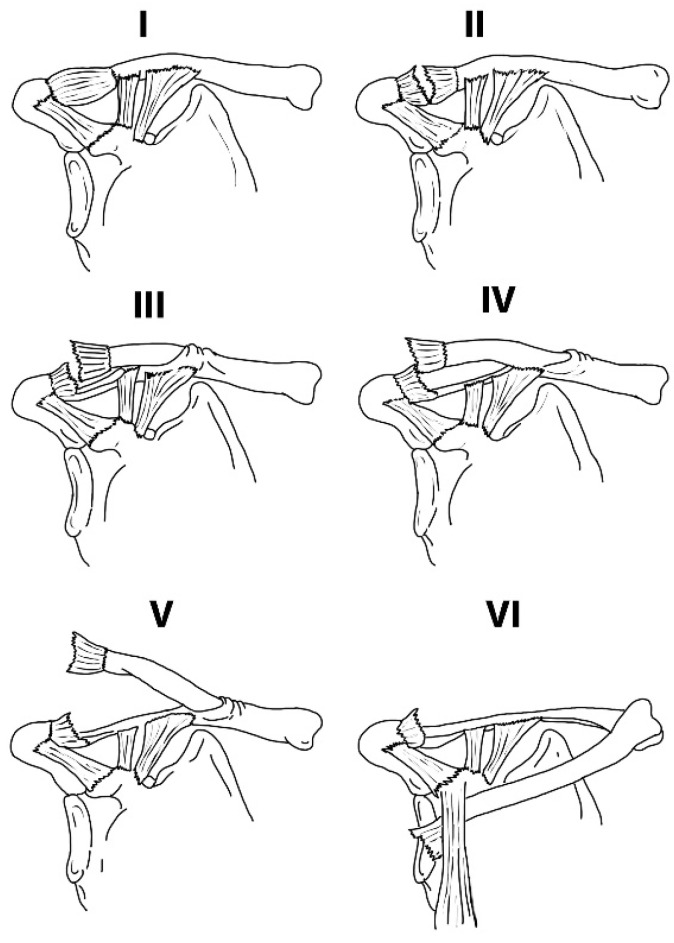
Dameron and Rockwood classification (Intact Coracoclavicular ligaments even in high-grade ACJ dislocation). Type I: Mild strain or sprain with intact periosteal sleeve; clinical instability may be absent. Type II: Partial tearing of dorsal periosteal sleeve, resulting in slight widening of lateral clavicular physis and mild clinical instability of the AC joint. Type III: Extensive disruption of the dorsal periosteal sleeve, leading to superior displacement of the distal clavicle; clinical instability present. Type IV: Comparable to Type III, but due to increased force, the distal clavicle is buttonholed through the trapezius muscle and fascia. Type V: Complete rupture of periosteal sleeve with superior displacement of the distal clavicle through the trapezius muscle into the subcutaneous tissue. Type VI: Subcoracoid displacement of the distal clavicle.The combination of an ACJ injury and other concurrent injuries can disrupt the Superior Shoulder Suspensory Complex (SSSC). Comprising bony and ligamentous elements like the glenoid, coracoid process, coracoclavicular ligaments, distal clavicle, ACJ and acromion, this complex stabilizes the shoulder, and thus its disruption can lead to the instability of the entire shoulder [11].

**Figure 2 jcm-12-05650-f002:**
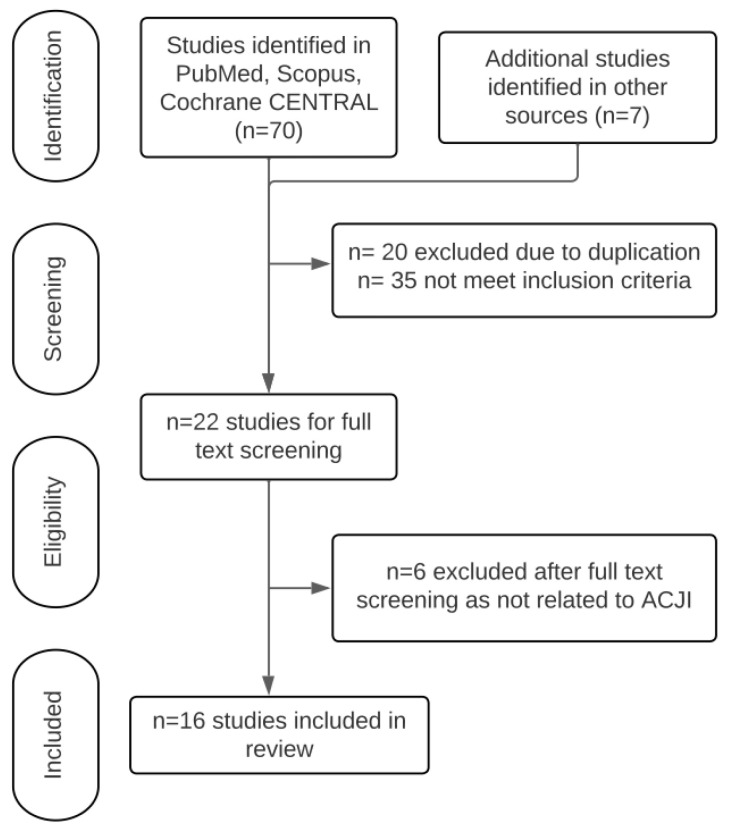
Study selection process. ACJI: Acromioclavicular joint injury.

**Figure 3 jcm-12-05650-f003:**
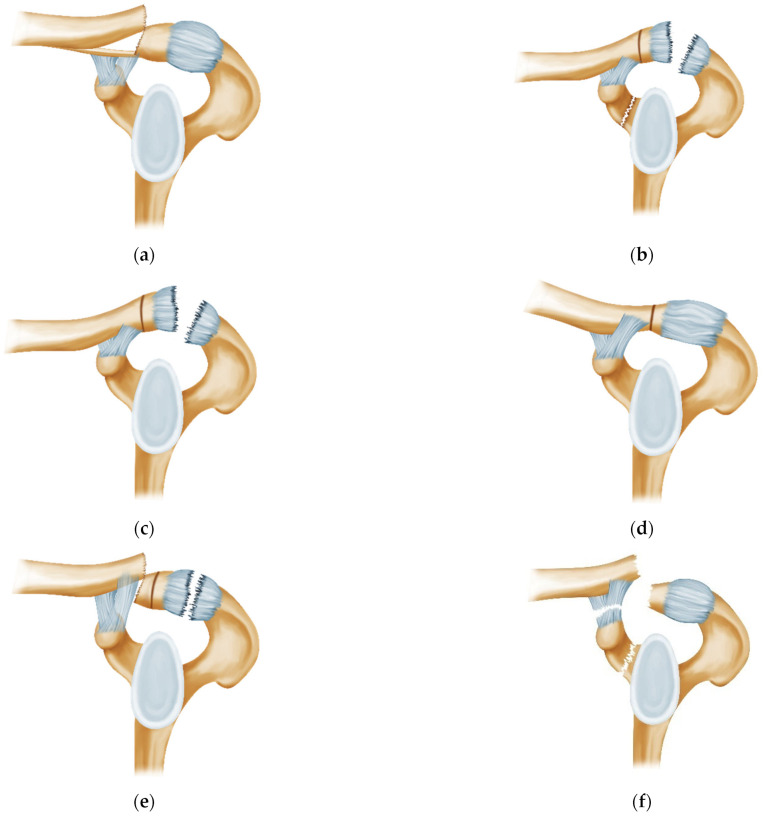
Categorization of ACJ injuries based on the site of injury of the SSSC and arranged in order of frequency of occurrence in the literature. The ACJ injury equivalents are all the types other than the isolated true ACJ dislocation. (**a**): Type 1—Pseudo-dislocation of the AC joint with intact CC ligaments. (**b**): Type 2—Coracoid fracture with AC joint dislocation. (**c**): Type 3—True dislocation of the AC joint. (**d**): Type 4—Voluntary atraumatic dislocation of the AC joint. The AC joint capsule is lax but intact. (**e**): Distal Clavicle fracture with AC joint dislocation. (**f**): Type 6—Triple Injury of the Superior Shoulder Suspensory Complex (Coracoid fracture, Coracoclavicular ligament rupture, AC joint pseudo-dislocation).

**Figure 4 jcm-12-05650-f004:**
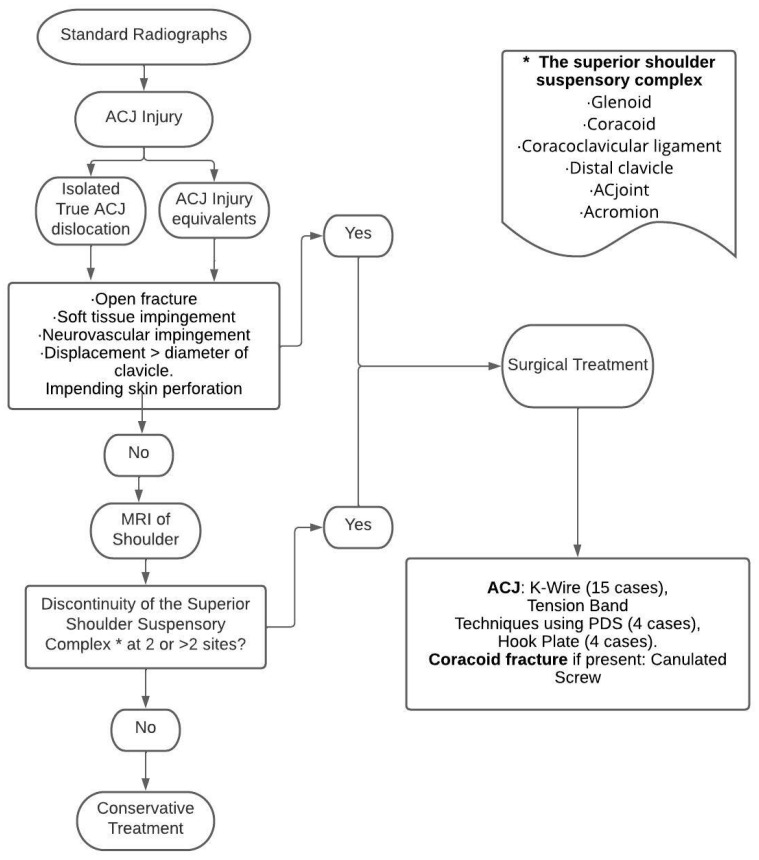
Management Algorithm based on the findings of this review. ACJ: Acromioclavicular joint. * Superior Shoulder Suspensory Complex: Glenoid, Coracoid, Coracoclavicular ligaments, distal Clavicle, Acromioclavicular (AC) joint, Acromion.

**Table 1 jcm-12-05650-t001:** Study characteristics and data extracted.

Study	Study Design	Participant Characteristics	Categories	Imaging	Intervention	Reason for Intervention	Outcome and Follow-Up (FU)
Rashid et al. [15] UK	Case series Retrospective analysis	*n* = 5, mean age = 12.8 years (9–14), 4 males, 1 female ACJ Pseudo-dislocation	Type 1	X-rays AP	ORIF (*n* = 4): Tension band technique using PDS Non-operative (*n* = 1)	High-grade injuries with displacement	Excellent clinical and radiological outcome. FU: 26 months
Richards and Howard [16] Canada	Case report	13-year-old male Distal Clavicle fracture ACJ Pseudo-dislocation	Type 1	X-ray—AP and Axillary views CT scan	Closed reduction under sedation	Soft tissue impingement	Return to competitive sports. FU: 12 weeks
Kirkos et al. [17] Greece	Case report	12-year-old male Distal clavicle fracture ACJ Pseudo-dislocation	Type 1	X-ray—AP view MRI	ORIF: 2 K-wires and nonabsorbable sutures	High-grade injury with suspected ligamentous disruption	Full ROM FU: 2 years
Kotb et al. [18] USA	Case report	11-year-old male Distal clavicle fracture ACJ Pseudo-dislocation	Type 1	X-ray AP view CT scan	ORIF: 3 K-wires	Clinical deformity High-grade injury	Return to full activity. FU: 8 weeks
Aebischer et al. [19] Australia	Case report	9-year-old male Distal clavicle fracture ACJ Pseudo-dislocation	Type 1	X-ray AP view	ORIF: 2 K-wires	Clinical and radiographic deformity	Return to sports. FU: 18 months
Goncalves et al. [20] Brazil	Case report	13-year-old girl. ACJ Pseudo-dislocation	Type 1	X-ray AP view of left shoulder	ORIF using K wire	Deformity, Soft tissue entrapment	Excellent outcome at FU of 3 months
Pedersen et al. [21] Germany	Case report	14-year-old male Coracoid fracture with ACJ dislocation	Type 2	CT scan MRI	Non-operative	Intact ACJ capsule, ligaments and CC ligaments	Return to sports at 12 weeks. FU: 5 months
Mondori et al. [22] Japan	Case series Retrospective analysis	9 coracoid fractures, *n* = 7 associated with ACJ dislocation in 6 males and 1 female, mean age = 14.5 years (11 to 17)	Type 2	X-ray and 3D- CT scan	ORIF (*n* = 3) Non operative (*n* = 4)	Unstable injury pattern. Conservative management performed only in patients who refused surgery	Excellent outcome. FU: 1 year
DiPaola and Marchetto [23] USA	Case report	15-year-old male with fracture of coracoid and ACJ dislocation	Type 2	AP, Lateral and Axillary X-ray of both shoulders	Non operative	Stable injury pattern with minimal displacement of coracoid	Excellent outcome. FU: 8 weeks
Kubiak and Slongo [24] Switzerland	Case series Retrospective analysis	15 clavicle fractures *n* = 1 patient (11-year-old male) had ACJ dislocation with clavicle fracture. *n* = 4 ACJ Pseudo-dislocation	Type 5 (*n* = 1) Type 1 (*n* = 4)	X-ray AP view	ORIF (*n* = 5) K-wire = 3 cases Osseous suture = 2 cases	Soft tissue /muscle impingement in 4 cases Shortening of shoulder girdle in 1 case	Individual FU and implant removal time not mentioned. Median FU for 15 patients was 88 days
Jettoo et al. [25] United Kingdom	Case report	12-year-old male Coracoid fracture with ACJ dislocation	Type 2	X-ray AP view CT scan	ORIF: 2 threaded half pins for ACJ and screw fixation for coracoid	Early mobilization	Full range of motion at 5 months, return to normal activities at 9 months
Kraus et al. [7] Germany	Case series Retrospective analysis 7 trauma centers	*n* = 7, mean age = 14.5 years (13 to 17), all males True ACJ dislocation	Type 3	Not mentioned	ORIF (*n* = 6): 4 Hook-plates and 2 K-wires. Arthroscopic procedure (*n* = 1): Tight rope	All high-grade injuries	No limitation of range of movement. FU: 6 weeks to 6 months
Sadeghi et al. [26] Netherlands	Case report	17-year-old female Voluntary dislocation of the ACJ	Type 4	X-ray, CT scan MRI arthrogram	Non-operative	Young age and minor complaints	Sub-optimum recovery at 1 year FU
Barchick et al. [27] USA	Case report	15-year-old female Voluntary dislocation of bilateral ACJ	Type 4	MRI Arthrogram right shoulder. No imaging for left	Arthroscopic augmentation of ACJ in right shoulder (*n* = 1) Non-operative for left (*n* = 1)	Pain and decreased function for right. No comment on left	Right: Poor outcome at 6 months Left: not mentioned
Acharya et al. [28] United Kingdom	Case report	12-year-old male Distal clavicle fracture with ACJ dislocation	Type 5	X-ray AP view CT scan	ORIF using k-wire	Unusual fracture pattern with unstable ACJ	Full functional recovery. Hypertrophic Scar FU: 1 year
Duerr et al. [29] USA	Case report	12-year-old male with ‘Triple Injury’. Coracoid fracture, CC ligaments avulsion and ACJ Pseudo-dislocation	Type 6	X-ray AP views of both shoulders	ORIF using Screw for coracoid, transosseous FiberWire sutures for ACJ and Suture Tak anchors (Arthrex) for CC ligaments	Triple injury with disruption of the SSSC	Excellent with return to sports at 1 year

ACJ: Acromioclavicular joint, CC: Coraco-clavicular, CT: Computerized Tomography, FU: Follow-up, MRI: Magnetic Resonance Imaging, ORIF: Open Reduction and Internal Fixation, PDS: Polydioxanone Sutures, SSSC: Superior Shoulder Suspensory Complex.

**Table 2 jcm-12-05650-t002:** Bias assessment through the Joanna Briggs Institute (JBI) critical appraisal checklist for case reports.

Checklist Questions	Richards Howards [16]	Acharya et al. [28]	Sadeghi et al. [26]	Barchick et al. [27]	Pedersen et al. [21]	Jettoo et al. [25]	Kirkos et al. [17]	Kotb et al. [18]	Aebischer et al. [19]	DiPaola and Marchetto [23]	Duerr et al. [29]	Goncalves et al. [29]
1.Were the patient’s demographic characteristics clearly described?	Yes	Yes	Yes	Yes	Yes	Yes	Yes	Yes	Yes	Yes	Yes	Yes
2.Was the patient’s history clearly described and presented as a timeline?	Yes	Unclear	Yes	Yes	Yes	Yes	Yes	Yes	Yes	Yes	Yes	Yes
3.Was the current clinical condition of the patient on presentation clearly described?	Yes	Unclear	Yes	Yes	Yes	Yes	Yes	Yes	Yes	Yes	Yes	Yes
4.Were diagnostic tests or assessment methods and the results clearly described?	Yes	Yes	Yes	Yes	Yes	Yes	Yes	Yes	Yes	Yes	Yes	Yes
5.Was the intervention(s) or treatment procedure(s) clearly described?	Yes	Yes	Yes	Yes	Yes	Yes	Yes	Yes	Yes	Yes	Yes	Yes
6. Was the post-intervention clinical condition clearly described?	Yes	Yes	Yes	Yes	Yes	Yes	Yes	Yes	Yes	Yes	Yes	Yes
7.Were adverse events (harms) or unanticipated events identified and described?	No	Yes	Yes	Yes	No	No	No	No	No	No	No	No
8. Does the case report provide takeaway lessons?	Yes	Yes	Yes	Yes	Yes	Yes	Yes	Yes	Yes	Yes	Yes	Yes

**Table 3 jcm-12-05650-t003:** Bias assessment through the Joanna Briggs Institute (JBI) critical appraisal checklist for case series.

Checklist Questions	Kraus et al. [7]	Kubiak and Slongo [24]	Mondori et al. [22]	Rashid et al. [15]
1. Were there clear criteria for inclusion in the case series?	Unclear	Yes	Yes	Yes
2. Was the condition measured in a standard, reliable way for all participants included in the series?	Yes	Yes	Yes	Yes
3. Were valid methods used for the identification of the condition for all participants included?	Yes	Yes	Yes	Yes
4. Did the case series have consecutive inclusion of participants?	Yes	Yes	Yes	Yes
5. Did the case series have complete inclusion of participants?	Yes	Yes	Yes	Yes
6. Was there clear reporting of the demographics of the participants in the study?	Yes	Yes	Yes	Yes
7. Was there clear reporting of the clinical information of the participants?	Yes	Yes	Yes	Yes
8. Were the outcomes or follow-up results of cases clearly reported?	Yes	Yes	Yes	Yes
9. Was there clear reporting of the presenting site(s)/clinic(s) demographic information?	Yes	Yes	Yes	Yes
10. Was the statistical analysis appropriate?	Yes	Yes	Yes	Yes

**Table 4 jcm-12-05650-t004:** Categorization of injuries around the AC joint in adolescents.

Types	Description	Disruption of the SSSC at	Total Cases	Operated
Type 1	Pseudo-dislocation of the AC joint	1 site (AC joint)	14	12
Type 2	Coracoid fracture with AC joint dislocation	2 sites (coracoid and AC joint)	10	4
Type 3	Isolated true dislocation of the AC joint	1 site (AC joint)	7	7
Type 4	Voluntary atraumatic dislocation	1 site (AC joint)	3	1
Type 5	Distal clavicle fracture with AC joint dislocation	2 sites (Clavicle and AC joint)	2	2
Type 6	Triple Injury (coracoid fracture, CC ligament rupture, AC joint dislocation)	3 sites (Coracoid, CC ligaments and AC joint)	1	1

AC: Acromioclavicular, CC: Coraco-clavicular, SSSC: Superior Shoulder Suspensory Complex.

## Data Availability

No new data other than the tables and figures already in the manuscript were created.

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
