# Peer review of "Acromioclavicular Joint Lesions in Adolescents—A Systematic Review and Treatment Guidelines"

_jcm, 2023, doi:10.3390/jcm12175650_

Round 1
Reviewer 1 Report
Thank you for the opportunity to review this work. The paper addresses an important topic. The comments I have made below are minor.
Introduction
The authors make the argument that AC injuries in pediatric patients are uncommon and that current treatment standards are not evidence-based. The authors start their argument with a quote that the injury is not of great concern due to the minor level of impairment associated with the injury. I suggest the authors strengthen their argument by adding some evidence that AC joint injury can lead to short and long-term impairments in pediatric patients.
Methods
The PICO question is appropriate. The authors might want to include a description of how they defined an ACJ injury for the purpose of their investigation. I am not sure why the authors excluded papers published prior to 2000.
Results
The authors might want to add a statement that they identified no RCT. The citations in table 2 do not seem to match, a couple of the numbers are repeated.
Discussion
I suggest the authors start their discussion with a summary statement that could be used as an answer to the PICO question that they sought to answer. I suggest the authors move their management algorithm earlier in the discussion, that seems to be what they were attempting to identify with their investigation. Finally, the authors might want to end their paper with some kind of statement regarding the need for a prospective study in the area.
Reviewer 2 Report
Review of the manuscript JCM-2523886
General comments
This paper is a systematic review that provides an overview of acromioclavicular joint injuries (ACJI) in adolescents. It analyzes 16 articles and categorizes the cases based on injury frequency and pattern.
The purpose of this study is to provide an organized overview of acromioclavicular joint injuries (ACJI) in adolescents and propose a management algorithm for these injuries. The study aims to analyze the existing literature on ACJI in children and adolescents, categorize the different types of injuries based on the site of injury to the Superior Shoulder Suspensory Complex (SSSC), and provide insights into the outcomes and management strategies for these injuries. The study also aims to address the limitations of the existing classification system and propose a new categorization system that can guide clinicians in the treatment of pediatric ACJI.
Title
“Superior Shoulder Suspensory Ligament Complex lesions in the adolescents – A systematic review and treatment guidelines” would be a more accurate title.
Introduction
Please provide an hypothesis
M&M
Line 71: "Janm, 2021”, please correct this.
Line 81: Please be clearer regarding who and how the management algorithm was developed.
Line 85: please be clearer regarding who were these two reviewers
RESULTS
Line 89: what about Cochrane Central Register of Controlled Trials?
Lines 136-138: “Rockwood’s classification was not considered important while making management decisions and could only be applied in 7 cases (isolated true AC joint dislocations) out of 32” Please provide references.
Line 152: Outcome and Complications – please provide the outcome measures used in each of the included studies. We suggest to incorporate this in Table 1.
Line 155: Please state how many papers reported on the existence of complications.
Line 158: Please just report results in this section. This should be moved to the discussion section: “Considering the low level of evidence and inhomogeneity of the data (population characteristics, diagnosis, imaging, intervention and outcome), formal statistical analysis and quantitative synthesis of data were not performed as even major differences could not reach significance.”
DISCUSSION
Please start the discussion with the most important finding of the study.
Lines 173-176: “Eidman in his classic 1980 presentation at the American Orthopaedic Society for Sports Medicine in Big Sky, Montana, described that true AC joint injuries classically do not occur in children below 13 years of age; only pseudo dislocations are seen below this age[8].” Please move this to the introduction section
Line 194: please move this illustration to the introduction section;
Lines 203-208: “The Superior Shoulder Suspensory Complex (SSSC) is formed by a bony and ligamentous ring formed by the glenoid, coracoid process, coracoclavicular ligaments, distal clavicle, ACJ, and the acromion that stabilizes the shoulder [28]. Rupture of this ring can lead to a decoupling of the upper extremity from the axial skeleton. Because of this disruption of the biomechanical ring, the transmission of cumulative forces from the axial skeleton is altered leading to altered mechanics and thereby decreased efficacy” Please move this to the introduction section.
Line 209: Please move this illustration to the results section;
Lines 218-280: This belongs in the results section. Please discuss the presented results in this section. We suggest that the authors maintain the discussion by sections, recalling the structure of the results.
It should be included in the discussion why do the authors believe that these injuries should be considered ACJ dislocations in the adolescent population when is considered a different entity in the adult population.
It should also be discussed in this section if the hypothesis of the study was proven or not and why.
CONCLUSION
The conclusion should be a short sentence that recalls the hypothesis of the study. Please avoid resuming the manuscript in this section.
Lines 295-305: “ACJ injuries in children and adolescents are rare, with a limited number of admissions each year. In this study, we employed Forscher's 1963 theory, as described in 'Chaos in the brickyard' [32] to filter out irrelevant studies and analyze the relevant ones. Using this approach, we developed a comprehensive treatment algorithm based on publishe reports and a categorization system that considers the pathoanatomy of the shoulder join This algorithm aims to provide clinicians with a practical framework for delivering appropriate care to patients with ACJ injuries. Additionally, we introduced the concept of "ACJ injury equivalents," which represents injuries with similar presentations and management principles. Similar to Monteggia equivalents in children, this concept is expected to evolve over time as it is supported by sporadic reports. By establishing this framework and terminology, we contribute to the ongoing understanding and management of ACJ injuries in this patient population.” This should be moved to the discussion section.
The quality of the English language is good
